# Impact of Isoquinoline Alkaloids on the Intestinal Barrier in a Colonic Model of *Campylobacter jejuni* Infection

**DOI:** 10.3390/ijms262110634

**Published:** 2025-10-31

**Authors:** Anna Duda-Madej, Przemysław Gagat, Jerzy Wiśniewski, Szymon Viscardi, Paweł Krzyżek

**Affiliations:** 1Department of Microbiology, Faculty of Medicine, Wroclaw Medical University, Chałubińskiego 4, 50-368 Wroclaw, Poland; pawel.krzyzek@umw.edu.pl; 2Faculty of Biotechnology, University of Wroclaw, Fryderyka Joliot-Curie 14a, 50-137 Wroclaw, Poland; przemyslaw.gagat@uwr.edu.pl; 3Department of Biochemistry, Molecular Biology and Biotechnology, Faculty of Chemistry, Wroclaw University of Science and Technology, 50-370 Wroclaw, Poland; jerzy.wisniewski@pwr.edu.pl; 4Faculty of Medicine, Wroclaw Medical University, Ludwika Pasteura 1, 50-367 Wroclaw, Poland; szymon.viscardi@student.umw.edu.pl

**Keywords:** berberine, biofilms, *Campylobacter jejuni* infections, chelerythrine, intestinal barrier, sanguinarine

## Abstract

Phytotherapy is a growing field of modern medicine, offering natural alternatives with multidirectional pharmacological effects. Among plant-derived bioactive compounds, isoquinoline alkaloids exhibit antioxidant, anti-inflammatory, and antimicrobial properties. Our in vitro model of campylobacteriosis confirmed that berberine reduces pathological changes in colonocytes not only through its direct antibacterial (minimum inhibitory concentration for pure berberine against *Campylobacter jejuni* was 64 μg/mL) and anti-biofilm (fourfold reduction in *C. jejuni* biomass) effects, but also through its protective effect on the morphostructure and secretory profile of host cells exposed to bacterial components. Furthermore, berberine stabilized intercellular junction proteins, modulated bile acid and arachidonic acid metabolism, and supported host-protective signaling pathways. These findings indicate that berberine acts through a dual mechanism—directly reducing bacterial virulence while enhancing intestinal barrier integrity and metabolic homeostasis. In summary, berberine appears to be a multifunctional phytochemical in the development of new strategies for the prevention and treatment of *C. jejuni*-induced gastrointestinal infections and epithelial barrier dysfunctions. The protective effect we have demonstrated may contribute to alleviating the phenomenon of “leaky gut,” commonly associated with campylobacteriosis.

## 1. Introduction

Invasive gastrointestinal infections (IGIs) represent the most acute form of all cases of gastrointestinal disorders. According to data from the World Health Organization (WHO), food-borne diseases remain a major global health problem. It is estimated that every year, nearly one in ten people worldwide falls ill due to unsafe food, with as many as 420,000 dying, including 125,000 children under the age of five [1]. Among these cases, IGIs, due to their pathomechanism of infection, are a particularly serious subgroup of intestinal diseases [2,3,4,5]. Pathogens causing them are characterized by a unique feature of penetrating epithelial cells of the small and/or large intestines. This leads not only to the development of diarrheas, but preeminently to tissue damage and, consequently, to inflammation [6,7]. IGIs are a serious problem in healthcare globally, being particularly dangerous in developing countries. This is related to poor sanitation and hygiene conditions. These factors, however, are not the only ones predisposing to IGIs. Weakened, defective functioning of the immune system is also an important contributor to the development of IGIs. Therefore, the incidence of these diseases most often affects children under the age of 5 and the elderly over the age of 65 [8].

Etiological agents of IGIs include different types of microorganisms. Among them, bacteria constitute the most prevalent group of pathogens and include *Campylobacter jejuni*, enteroinvasive *Escherichia coli* (EIEC), *Listeria monocytogenes*, *Salmonella* spp., *Shigella* spp., and *Yersinia enterocolitica.* Predominantly, these species are capable of synthesizing invasins—specialized microbial proteins that facilitate traversing host tissue barriers and enable evasion of host immune defenses [9,10]. Despite the heterogeneity of pathogenic mechanisms employed by these bacterial species, the resulting complications are often severe, protracted, and clinically significant. Although they mostly affect the gastrointestinal tract, some of them also include potentially life-threatening or permanently debilitating extraintestinal complications, such as: (i) hemolytic uremic syndrome (HUS); (ii) meningitis; (iii) Reiter’s syndrome (reactive arthritis); (iv) osteomyelitis; and (v) sepsis [11].

Among bacterial pathogens associated with IGIs, *C. jejuni* warrants particular attention [12]. Given its low infectious dose—estimated at merely 500 to 800 bacterial cells [13,14]—this “stealthy invader” is frequently underrecognized and overlooked in clinical diagnostics. Several factors contribute to this diagnostic gap, including: (i) prolonged environmental persistence with survival up to 24 h on kitchen surfaces and several weeks on chilled meat products [15]; (ii) delayed onset of clinical symptoms [16]; (iii) repeat dietary exposure to subinfectious doses of this bacterium [17]; (iv) delayed manifestation of post-infectious complications [18]; (v) symptomatic similarity to other conditions (i.e., acute appendicitis) [19]; and (vi) a frequent underestimation and misclassification of cases as “self-limiting” [20]. The pathogenicity of *C. jejuni* is multifactorial and involves a comprehensive array of virulence determinants operating at all stages of infection, including motility, adhesion, invasion, oxidative stress resistance, and biofilm formation [21].

This multifaceted pathogenicity of *C. jejuni* contributes not only to gastrointestinal complications, but also to extraintestinal complications. Delayed consequences of *C. jejuni* infection include a vast list of serious and life-threatening ailments, such as: (i) post-infectious irritable bowel syndrome (PI-IBS) [22,23]; (ii) inflammatory bowel disease (IBD) [24]; (iii) Guillan-Barré syndrome (GBS) [25,26]; (iv) reactive arthritis [22]; and (v) development of colorectal cancers [27,28], their metastases [29], and mucosa-associated lymphoid tissue (MALT) lymphomas of the small intestine [30]. The participation of *C. jejuni* in the development of these diseases is undoubtedly promoted by the biofilm-forming ability of this bacterium. A biofilm is a multicellular structure of microorganisms surrounded by a thick layer of extracellular matrix that limits the effectiveness of the immune system and antimicrobial therapies, allowing them to cause chronic infections. Among *Campylobacter* strains, a crucial role of flagellin-related genes (e.g., *flaA*, *flaB*, *flaC*, *flag*, *fliA*, *fliS*, *flhA*) in the initiation of early adhesion and maintenance of the biofilm structure was demonstrated [31]. Additionally, genes involved in shaping the biofilm’s composition (e.g., *waaF*, *lgtF* and *EptC*) [21] and stabilizing its matrix (e.g., *spot*, *ppk1/2*—stringent response regulators) [32] have been identified. A range of antioxidant genes, including, e.g., *ahpC*, *katA*, and *sodB*, have also been shown to enhance biofilm resilience under environmental stress [21]. Recent studies highlight the role of *cadF* not only in facilitating enterocyte invasion via activation of the MAPK/ERK (microtubule-associated protein kinase/extracellular regulated kinase) signaling pathway [33], but also in promoting initial adhesion during biofilm formation [32]. Taking this into account, the biofilm created by *C. jejuni*, on the one hand, enables the persistence of this bacterium on the surface of the intestinal epithelium, while on the other hand, enables the long-lasting, destructive deposition of lytic factors that negatively affect the condition of this organ [34]. *C. jejuni* has been shown to be involved in each stage of biofilm formation: initiation of early adhesion [31], composition modeling [21], and matrix stabilization [32]. In addition, a number of antioxidative proteins by *C. jejuni* have been detected, all of which determine the biofilm’s tolerance to environmental stress and its stabilization [21].

Importantly, it has been proven that the consequences of *C. jejuni* infection can be very serious and life-threatening. They can contribute to colorectal carcinogenesis and metastasis. This oncogenic potential has been associated with strains expressing cytolethal distending toxin (CDT), which can activate: (i) the JAK2 (Janus kinase 2)/STAT3 (signal transducer and activator of transcription 3)/MMP9 (matrix metalloproteinase-9) signaling axis implicated in tumor development [27,28] and (ii) glycogen synthase kinase 3 beta (GSK3b), a key regulator of cancer metastasis [29]. Moreover, although a direct causal association has not been definitively established, *C. jejuni* infections have been implicated in immunoproliferative small intestinal disease (IPSID), also known as alpha heavy chain disease. Studies have demonstrated that a strong mucosal IgA response induced by chronic *C. jejuni* infection can lead to persistent stimulation of the mucosal immune system and aberrant secretion of truncated α-heavy chains. This process is thought to contribute to the pathogenesis of mucosa-associated lymphoid tissue (MALT) lymphoma involving the small intestine [30]. The role of *Campylobacter* spp. in invasive gastrointestinal infections is shown in Figure 1.

Through many years of different research, it has been demonstrated that *C. jejuni* infection contributes to the disruption of the intestinal barrier integrity, which serves as a critical line of defense against environmental agents. The pathogen-secreted proteases and toxins (i.e., HtrA) induce proteolytic cleavage of essential tight junction (TJ) proteins [35,36,37]. This structural degradation compromises the epithelial monolayer and facilitates the translocation of pathogens and their virulence factors into the subepithelial space. Furthermore, in vitro studies have shown that *C. jejuni* infection alters the intracellular localization of TJ-associated proteins, e.g., occludin, resulting in their absence at the apical membrane. These changes lead to a marked impairment of the epithelial barrier function and an increased paracellular permeability [38]. As a consequence, a complex cascade of immunometabolic responses is activated in the host, leading to the development of various acute and chronic clinical complications. Figure 2 shows a simplified schematic of the pathogenesis of *C. jejuni* and the consequences of this infection.

The evolutionary adaptability and the genetic diversity of *C. jejuni* strains allow for the widespread transfer of resistance genes and constitute a major defensive strategy against antimicrobial therapies. In the context of the global antibiotic crisis and the remarkable adaptability of *C. jejuni*, support from natural sources appears to be increasingly necessary. Naturally occurring compounds, often ingested through the diet, may assist in combating invasive bacterial pathogens either directly—by their antimicrobial activity, disrupting cell membrane, and increasing their permeability—or indirectly, by weakening microbial defense mechanisms, e.g., biofilm. Isoquinoline alkaloids found in *Chelidonium majus* L. (sanguinarine (SAN; Figure 3A), chelerythrine (CHE; Figure 3B), and berberine (BBR; Figure 3C) constitute very promising compounds with a variety of health-promoting properties, e.g., antimicrobial [39,40,41,42], anti-biofilm [43,44,45,46], anti-inflammatory [47,48,49,50], and anticancer [51,52,53,54]. Furthermore, BBR has been shown to modulate the gut microbiota and enhance intestinal barrier function by upregulating the expression of TJ proteins, e.g., ZO-1, occludin, claudin-1, and claudin-4 [55,56,57,58].

In this article, we present the results highlighting the protective effect of isoquinoline alkaloids from *Ch. majus* L. on colonocyte cells using an in vitro model of *C. jejuni* infection. Considering its most favorable cytotoxicity profile towards colonocytes, the main body of experiments was focused solely on BBR. Using culture and microscopic methods, supported additionally by metabolomic data, a potent anti-biofilm activity against *C. jejuni* and an intestinal barrier-stabilizing effect of BBR was proved. The activity of BBR in invasive gastrointestinal infections is shown in Figure 4.

## 2. Results

The MIC values for CHE, BBR, and SAN against *C. jejuni* were tested in the range of 1–128 µg/mL in three selected media, i.e., BHI (a reference medium for non-fastidious bacteria), BHI5 (a reference medium for fastidious bacteria), and ACF10 media (a colon-mimicking artificial medium). Turbidity in 48-well plates was read visually and then additionally confirmed with fluorescent observations of the bacterial biomass (Appendix A). The obtained MIC values are presented in Table 1.

Although lower MIC values were observed for SAN and CHE, according to the cytotoxicity assays performed on the CCD841 CoN cell line, only BBR was selected for further analysis, as it exhibited no adverse effects on the tested colonocytes, with a mean viability of approximately ~94–101% (Appendix A) across all evaluated concentrations (½ × MIC, MIC, and 2× MIC) and no dose effect (Wilcoxon, *p*_(Bonf)_ ≥ 0.086, Appendix A). In contrast, CHE and SAN were significantly cytotoxic at all doses, with mean viabilities of ~22–32% (CHE) and ~24–25% (SAN), respectively, and likewise no within-compound dose differences (Wilcoxon, *p*_(Bonf)_ ≥ 0.088, Appendix A). When data were pooled across concentrations (n = 12 per compound), BBR yielded higher viability than both CHE and SAN (Wilcoxon, *p*_(Bonf)_ = 1.09 × 10^−4^ and 1.08 × 10^−4^, Appendix A), whereas CHE vs. SAN did not differ (Wilcoxon, *p*_(Bonf)_ = 1, Appendix A). The survival rate of the CCD841 CoN cell line in the presence of the tested compounds BBR, CHE, and SAN is shown in Figure 5.

To demonstrate the protective role of BBR in the in vitro model of *C. jejuni* infection, we first examined its ability to influence the biofilm formed by the tested strain. In the initial stage of assessing anti-biofilm activity, we determined the effect of the applied culture medium. For this purpose, we tested three variants of media, i.e., BHI5 (a reference microbiological medium), EMEM10 (a reference cell line medium), and E/Amix (a colon-mimicking artificial medium) (Figure 6). Regardless of the culture conditions, bacteria treated with the MIC of BBR exhibited a significant, approximately 4-fold reduction in biofilm formation (Figure 6; Wilcoxon, *p*_(Bonf)_ ≤ 0.000133, Appendix A). As expected, BHI5 promoted the highest biofilm production, while the difference in the biofilm formation between EMEM10 and E/Amix media was insignificant (*p*_(Bonf)_ = 1.00, Appendix A). Irrespective of the culture medium applied, BBR consistently and strongly reduced biofilm formation of *C. jejuni* compared to its counterpart without BBR: BHI5 vs. BHI5 + BBR, EMEM10 vs. EMEM10 + BBR, and E/Amix vs. E/Amix + BBR (*p*_(Bonf)_ ≤ 1.11 × 10^−5^, Appendix A). Taking into account the similar level of BBR activity in anti-biofilm activity, we decided to conduct further experiments in a medium resembling the host environment (E/Amix).

To obtain a more precise picture of the effect of BBR on the morphostructure of the biofilm produced in the E/Amix medium, we performed a post-microscopic analysis of the size distribution of all biofilm-like aggregates attached to the titers’ wells. We showed that BBR treatment significantly shifted the biofilm aggregates towards more dispersed (87.2% as small- and 12.8% as medium-sized; Table 2) compared to the control samples (40.8%, 46.5%, 9%, and 3.7% categorized as small, medium, big, and large, respectively; Table 2). Accordingly, BBR presents anti-biofilm activity against *C. jejuni* both by directly reducing the biomass of this structure and by decreasing the degree of bacterial autoaggregation.

During the microbiological experiments, the E/Amix medium was selected as optimal for the growth of the tested *C. jejuni* strain, while maintaining its host-mimicking properties. Therefore, before proceeding to studies on the CCD841 CoN cell line, we performed a comparative analysis of its growth in two media: EMEM10 (a reference cell line medium) and E/Amix (a colon-mimicking artificial medium). As shown in Figure 7, the morphology and adhesive capacity of colonocytes were similar in both media. Consistent with our primary goal of reproducing host-like conditions, subsequent experiments were conducted using E/Amix.

Next, we examined whether the MIC of BBR has a protective effect on colonocytes exposed to the *C. jejuni* post-culture supernatant. For this purpose, a time-dependent analysis of the surface coverage of the CCD841 CoN cell line grown on semipermeable membranes was performed. We found that exposure to the bacterial post-culture supernatant contributes to a decrease in host cell density and that co-treatment with BBR preserves the monolayer (Figure 8A). The ranking of medians suggested a pattern for the four groups: control/BBR ≥ BBRCamp > Camp for all times investigated but 24 h; however, the differences among the groups became statistically significant by 72 h and 96 h (Kruskal–Wallis, *p* < 0.05, Appendix A); the reduction at 96 h was as high as 63.5–72.4% (Figure 8B).

The percentage coverage of the membrane surface with CCD841 CoN cells changed at 72 and 96 h of an in vitro infection model (*p* = 0.0444 and *p* = 0.0329, respectively). The lowest coverage was observed in the group containing *C. jejuni* supernatant (Camp), followed by samples exposed to BBR (BBRCamp), and the highest coverage was seen in the control and BBR groups. We have therefore demonstrated that exposure of colonocytes to *C. jejuni* supernatant reduces their density, and that simultaneous treatment with BBR partially protects the monolayer compared to infection alone.

To gain a broader insight into the protective effect of BBR on colonocyte cells, we performed a microscopic examination of their size distribution as a measure of physiological status. We observed that on the fourth day of incubation, control samples, BBR-treated samples (BBR), and samples exposed to both BBR and *C. jejuni* post-culture supernatant (BBRCamp) had similar dimensions (Wilcoxon, *p*_(Bonf)_ > 0.05, Appendix A; Figure 8C), whereas cells exposed only to *C. jejuni* post-culture supernatant (Camp) showed almost a twofold reduction in their dimensions (Wilcoxon, *p*_(Bonf)_ < 0.05, Appendix A; Figure 8C) and a characteristic elongated sickle shape (Figure 8D). Concisely, it was shown that BBR has a strong protective effect on intestinal cells against toxic metabolites/virulence factors of *C. jejuni,* both by keeping their physiological growth and cellular morphology.

The Lucifer yellow assay confirmed that *C. jejuni* significantly weakens tight junctions (TJ) between colonocytes. In our studies, we observed that the tested strain affected TJ already on the first day of the in vitro infection model. The impact on colonocytes increased slightly over time, ranging from 13.8 ± 0.21% at 48 h to 15.1 ± 0.29% at 96 h (Figure 9, Appendix A). In contrast, BBR showed a permeability level of approximately 11% throughout the experiment, comparable to that in the untreated control. Notably, BBR alone had a protective effect on tight junctions compared to the control (Wilcoxon, *p*_(Bonf)_ < 0.05, Appendix A), while the combination of BBR and *C. jejuni* supernatant (BBRCamp) is consistently lower than the *C. jejuni* supernatant alone (Camp; Wilcoxon, *p*_(Bonf)_ < 0.05, Appendix A) and became indistinguishable from the control at 72–96 h. This indicates a protective, intestinal barrier-stabilizing effect of BBR in an in vitro model of campylobacteriosis.

In the final stage of the experiments, we aimed to assess the effect of BBR on metabolic changes in colonocytes exposed to *C. jejuni* post-culture supernatant. By comparing the “metabolic fingerprints” of colonocytes under different conditions, we observed several response patterns to these stimuli. A Kruskal–Wallis test applied per metabolite across 12 groups: control, BBR, Camp (*C. jejuni* supernatant), and BBRCamp (BBR + *C. jejuni* supernatant) at 0/48/96 h, with Benjamini–Hochberg FDR correction, yielded 47 significant metabolites (*p* < 0.05), related to either primary or secondary metabolic pathways (Appendix A). Among them, the following classes were detected: amino acids and their derivatives, lipids and phospholipids, bile acids and conjugates, phenolic and aromatic compounds, and energy compounds (Figure 10).

Secretory profile analysis allowed for the identification of three independent groups: (1) control and BBR at 0/48 h, (2) *C. jejuni* supernatant alone (Camp) and with the addition of BBR (BBR + *C. jejuni* supernatant) at 0/48 h, and (3) all 96-h samples in which control and BBR separated from Camp and BBRCamp. These results clearly indicate that time amplifies the differences between the secreted metabolites, while *C. jejuni* supernatant (Camp) remains a visible conditioning factor. Furthermore, simultaneous exposure of colonocytes to both the post-culture supernatant of *C. jejuni* and BBR (BBRCamp) contributed to the reduction of metabolic changes induced by *C. jejuni* supernatant components (Camp).

## 3. Discussion

Phytotherapy is increasingly regarded as a safer, natural alternative to conventional pharmaceuticals [62]. With very limited side effects and no impact on generating antibiotic resistance, it is undoubtedly gaining attention, with a noticeable integration of traditional medical systems (e.g., traditional Chinese medicine, European folk medicine) into modern healthcare [7,63]. The dynamic development of phytotherapy is driven by the pharmacologically documented, multidirectional effects of various natural preparations. Moreover, the easy access to phytotherapeutic preparations contributes to their popularity and widespread use. This is particularly important in the context of the growing prevalence of persistent diseases, e.g., diabetes, cardiovascular diseases, and chronic inflammatory states [64,65,66]. All of this translates into a growing recognition that plant-derived agents are considered not only as supplements to standard, long-term pharmacotherapies, but also as important components of preventive healthcare strategies [67,68].

One of the most important groups of bioactive compounds found in plants is alkaloids, substances with a broad spectrum of biological activities. One of the best-known among them is BBR, an isoquinoline alkaloid with proven multidirectional health-promoting properties. The high therapeutic potential of this compound is driven by its antioxidant, anti-inflammatory [69], anti-neurodegenerative [70,71], anticancer [72,73], and antimicrobial [74,75] activity. Although research on the activity of natural compounds against microorganisms involved in various pathogenic processes is currently experiencing a significant renaissance, studies on the activity of BBR against *Campylobacter* spp. are very scarce in the literature. In one of the few available reports, Fukamachi et al. demonstrated the beneficial effect of BBR on this invasive pathogen. The authors tested a Hangeshashinto extract, containing BBR among 15 potentially active components, against *C. jejuni* ATCC 29428 and showed its antibacterial activity [10,76]. However, the extent to which this activity was attributed by BBR is unclear. To the best of our knowledge, the experiments presented in the current study are the first to determine the MIC value for pure BBR in vitro against *Campylobacter* spp. We observed that this alkaloid inhibits the growth of *C. jejuni* ATCC 81-176 at a concentration of 64 μg/mL, confirming its promising antibacterial potential and opening avenues for future in vivo investigations.

Several decades of research on biofilms has proven their great clinical significance in the medical sector [77]. In addition to their well-documented protective function against numerous unfavorable environmental factors, such as antimicrobial agents, components of the immune system, and detergents, biofilms also exert direct toxicity towards host cells [78]. The cytotoxicity of biofilms is strictly linked to the powerful sorption properties of the matrix and its propensity to accumulate high concentrations of virulence factors with lytic activity [78,79]. This phenomenon is even more significant in the case of biofilms produced by microorganisms capable of chronically infecting the host, including *C. jejuni* [20]. There are numerous scientific reports indicating the crucial role of this bacterium’s biofilm in both survival outside the host and the exacerbation of disease pathologies once it enters the host [80]. Therefore, the search for anti-biofilm substances against *C. jejuni* as a strategy supporting therapeutic success and limiting the pathogenicity of this bacterium is recommended [81].

In the current article, we investigated the impact of BBR on the biofilm-forming capacity of *C. jejuni* and demonstrated very promising properties of this compound. MIC values of BBR not only contributed to a four-fold reduction in the total biofilm biomass formed under the research conditions, but also promoted a more dispersed phenotype of *C. jejuni* cells adhering to abiotic surfaces. Our observations indicating the high anti-biofilm potential of BBR are consistent with previous reports focusing on other microorganisms [82]. In light of the data we provide, BBR, by serving as a substance maintaining microorganisms in their planktonic phase, could potentially be applied in combination with antibiotics, facilitating better antibiotic penetration and reducing biofilm-dependent tolerance to antimicrobials.

The above-mentioned data on the anti-virulence potential of BBR justifies further studies, as current evidence indicates that *C. jejuni* damages the intestines through several mechanisms, including adhesion and biofilm formation, invasion, and production of toxins, all of which induce a strong inflammatory response [35,83]. These processes lead to the destruction of epithelial cells, microvilli atrophy, and the formation of mucosal ulcers [84,85]. The intestinal barrier in the course of *C. jejuni* infection can be damaged in two ways: (1) disruption of TJ [86,87] and (2) redistribution of barrier proteins [88]. The key factor in the first pathway is the serine protease HtrA, secreted by the bacterium into the environment [86,87]. It cleaves key proteins, including occludin, contributing to increased paracellular permeability [38]. The second pathway involves the relocation of major barrier proteins (e.g., occludin, claudin-8, and E-cadherin) from the TJ area into the cell interior, resulting in a weakening of the junction structure [88,89]. It has been shown that the effect of intestinal barrier disruption and increased intestinal mucosal permeability (irrespective of the pathway) is dose-dependent [90]. These in vivo data correlate well with our current in vitro findings. Our experiments with supernatant from *C. jejuni* cultures confirm that the damage to the intestinal barrier develops over time and increases with the duration of infection [91,92]. We discovered that the highest degree of disruption of intercellular connections in colonocytes occurred from day 1 post-infection, while marked relocation of occludin to the cytoplasm was observed by day 4 (Appendix A). These *C. jejuni*-induced pathological alterations in the morphostructure of colonocytes are consistent with the occurrence of clinical symptoms in campylobacteriosis, which typically appear 2–5 days after infection, as reported by the WHO [93]. Our results also align with previous in vitro and in vivo studies demonstrating that the decrease in transepithelial electrical resistance (TER) during *Campylobacter* infection leads to the development of “leaky gut syndrome” [88], which subsequently promotes inflammation and increases susceptibility to secondary gastrointestinal infections.

BBR has been shown to modulate gut microbiota composition [94], enhance intestinal barrier function (e.g., by upregulating junctional proteins such as ZO-1 and occludin) [95,96], and exert strong anti-inflammatory effects [97,98]. In our in vitro model of campylobacteriosis, the maximal protective effect of BBR on junctional integrity was observed on days 3 and 4 of CCD841 CoN cell infection. Taken together, these findings suggest that BBR may be of particular interest for mitigating *C. jejuni*-induced intestinal pathology and reducing its pathogenic potential.

Previous studies have demonstrated that BBR modulates bile acid metabolism, influencing gut microbiota composition and limiting the growth of pathogenic bacteria [99,100,101]. It has been shown that BBR increases the levels of primary bile acids and their conjugates, while reducing secondary ones. These changes lead to the activation of the FXR signaling pathway (farnesoid X receptor, a nuclear receptor activated by bile acids) and to decreased activity of bile salt hydrolases (BSHs), enzymes essential for bacterial survival in the intestinal lumen [102,103]. In our in vitro model, the use of BBR enabled the evaluation of its alleviating properties in colonocytes exposed to the *C. jejuni* post-culture supernatant containing metabolites and virulence factors secreted by this pathogen. We confirmed that BBR significantly influenced the pool of bile acids and their derivatives (e.g., cholate, tauro-/glyko-conjugates: taurochemodesoxycholic acid, glycocholate, tauroallocholic acid) secreted by colonocytes. This effect indicates enhanced environmental stress and impaired physiological detoxification mechanisms of pathogens within the intestinal milieu. Under control conditions, *C. jejuni* supernatant induced characteristic metabolic changes, including reduced levels of specific amino acids and lipids, along with the presence of metabolites associated with bacterial adaptive mechanisms. In contrast, in the presence of BBR, a clear shift was observed. A particularly noteworthy phenomenon was detected in the level of arachidonic acid produced by colonocytes treated with BBR and *C. jejuni supernatant*, suggesting an activation of host protective signaling pathways. Moreover, these findings indicate that BBR may restrict pathogen survival by modulating the metabolic environment while reinforcing host protective functions, facilitating pathogen elimination. This is pointing to a potential anti-*Campylobacter* activity of BBR during infection. Our results in this regard are consistent with previous reports describing BBR as a modulator of metabolic pathways linked to arachidonic acid [104,105].

Despite the promising results obtained in this study, it is important to consider the pharmacokinetic and pharmacodynamic limitations of BBR. Following oral administration, BBR exhibits low bioavailability due to extensive first-pass metabolism. Zuo et al. (2006) demonstrated that the liver is the primary site of BBR biotransformation, with major metabolites and glucuronide conjugates detected in hepatic tissue and bile within 0.5 and 1 h after administration, respectively [106]. Although the gut microbiota of germ-free rats (administered 40 mg/kg BBR) showed limited direct metabolic activity toward BBR, it significantly influenced the enterohepatic circulation of metabolites [106].

Therefore, the development of alternative delivery strategies and methods of administration is crucial for improving BBR absorption and therapeutic efficacy. Nanocarriers, liposomal formulations, and microfluidic systems have shown encouraging results in improving BBR solubility, stability, and intestinal absorption [107,108,109]. Furthermore, structural modifications of BBR have improved pharmacological performance while reducing adverse effects [110,111]. Recent studies have demonstrated that lactoferrin-based nanoparticles loaded with BBR and SAN, combined with conventional antibiotics, significantly enhance intracellular drug delivery and antibacterial activity compared to free drug forms [112]. This highlights the potential of a nanocarrier-mediated targeting system to overcome intracellular sequestration of pathogens and improve the therapeutic outcomes of poorly bioavailable alkaloids. Future research employing flow-based pharmacokinetic models, including hepatic cell systems and organ-on-a-chip technologies, will be crucial for elucidating BBR metabolism, toxicity, and tissue accumulation [113,114]. These approaches will support the establishment of standardized dosing, optimized delivery routes, and defined treatment durations for clinical use. The present findings provide new insights into the antimicrobial, anti-biofilm, and intestinal barrier-modulating potential of BBR and open avenues for future in vivo investigations.

## 4. Materials and Methods

### 4.1. Compounds Tested and Culture Media

#### 4.1.1. The Tested Compounds Were Purchased from Chemland (Zgierz, Poland)

According to the data provided by the manufacturer, they presented the following purity (HPLC): berberine, BBR (5,6-dihydo-9,10-dimethoxybenzo[g]-1,3-benzodioxdo[5,6-a] qlinolizinium): 97%; chelerythrine, CHE (1,2-dimethoxy-12-methyl-[1,3]dioxolo [4′,5′:4,5]benzo[1,2-c]phenanthridin-12-ium): 98%; and sanguinarine, SAN (13-methyl-[1,3]benzodioxdo[5,6-c]-1,3-dioxdo[4,5-i]phenanthridinium): min. 95%.

#### 4.1.2. LC-MS Analysis of Metabolomics Samples

The LC-MS grade water, methanol (MeOH), acetonitrile (ACN), and formic acid (FA)—Witko; Poland. The Protein LoBind microcentrifuge tubes (1.5 mL)—Eppendorf; Barkhausenweg, Germany. Leucine enkephalin (Waters, Milford, MA, USA) was utilized as a LockMass. Deuterated Amino Acid Standard Mixture—Merck, Darmstadt, Germany.

#### 4.1.3. Bacteriological Media

Brain Heart Infusion (BHI, Oxoid, Pol-Aura, Zawroty, Poland; see for details Appendix A) broth with 5% fetal calf serum (FCS, Ebsdorfergrund, Hesja, Germany), referred to for publication purposes as **BHI5**. Columbia agar media (CA, Oxoid, Pol-Aura, Poland; see for details Appendix A) containing 5% hemolyzed blood (horse blood defibrinated; TCS Biosciences, Buckingham, UK), referred to for publication purposes as **CA5**.

#### 4.1.4. Cell Line Media

Eagle’s Minimum Essential Medium (EMEM, ATCC, Manassas, VA, USA, State of Virginia; see Appendix A for details) supplemented with 10% fetal bovine serum (FBS, Biowest, Nuaillé, France) and antibiotic solution (Merck, Darmstadt, Germany), referred to for publication purposes as **EMEM10**. Artificial Colonic Fluid (ACF, Biochemazone, Canada; for details, see Appendix A) supplemented with 10% FCS, referred to for publication purposes as **ACF10**. The mixture of EMEM10 and ACF containing 20% FCS (1:1, *v*/*v*), referred to for publication purposes as the **E/Amix**.

### 4.2. Antimicrobial Experiments

#### 4.2.1. Revival of the Strain

The *Campylobacter jejuni* ATCC 81-176 strain (BAA 2151) was obtained from the American Type Culture Collection. It was kept frozen at −80 °C until the study. It was taken through the process of revival in BHI5 (for details see Section 4.1.3) and incubation under microaerophilic conditions (5% O_2_, 15% CO_2_, 80% N_2_; generated by Genbox microaer kits, BioMérieux, Warszawa, Poland) at 37 °C. The strain was then transferred to CA5 (for details see Section 4.1.3) and cultured for 3–5 days under similar environmental conditions.

#### 4.2.2. Experimental Evaluation of the Culture Conditions

To determine the optimal supplementation conditions for subsequent experiments, the effect of FCS on the growth of the *C. jejuni* was initially evaluated in two media: a standard BHI and a host-mimicking ACF. Bacterial cultures at 10^7^ CFU/mL were prepared in each medium and supplemented with FCS at final concentrations of 2.5%, 5%, and 10%. Based on the growth performance and the absence of noticeable inhibitory effects, BHI5 (for details see Section 4.1.3) and ACF10 (for details see Section 4.1.4) were selected for further analysis.

#### 4.2.3. Activity Against Planktonic Cells

The minimum inhibitory concentrations (MICs) of BBR, SAN, and CHE against *C. jejuni* were estimated using a microdilution assay in 24-well titer plates (Bionovo, Legnica, Poland). A series of dilutions in the range of 1–128 µg/mL in a final volume of 500 µL was made from the stock solution of the tested compounds prepared in DMSO (Chempur, Piekary Śląskie, Poland). To each well, a suspension of the tested strain at a final density of 10^7^ CFU/mL previously cultured in BHI5 liquid medium for 24 h under microaerophilic conditions was added. The experiments were conducted in three different culture media: BHI5 for all three compounds, and also EMEM10 (for details see Section 4.1.4) and E/Amix (for details see Section 4.1.4) for BBR only. Wells containing bacterial suspensions without tested compounds served as positive controls, while wells with sterile culture media acted as negative controls. The plates were incubated for 3 days at 37 °C under microaerophilic conditions and with shaking at 100 rpm (MaxQ 6000, Thermo Fisher, Waltham, MA, USA). The MIC was defined as the lowest BER, SAN, and CHE concentration that prevented visible turbidity in the well. Each test was carried out in four biological replicates (n = 4).

#### 4.2.4. Activity Against the Biofilm

After finishing tests determining activity against planktonic cells, from each well of the titer plate marked as MIC (a tested sample) and not containing BBR (a control sample), the entire bacterial suspension was removed and gently washed once with a PBS solution. Then, rabbit polyclonal anti-*C. jejuni* antibodies conjugated to FITC (dilution 1:10; ThermoFisher, Waltham, MA, USA) were added to each well in order to stain the biomass biofilm. The plate was incubated in the dark and at room temperature for 15 min, and then the samples were observed using a Carl Zeiss inverted fluorescence microscope (GmbH, Jena, Germany). The degree of area occupation by adherent bacterial cells was calculated using a Bioflux Montage software version 7.10.3.390 (Fluxion, San Francisco, CA, USA). To minimize the background fluorescence level, a 10% increase in the contrast was equally applied to all analyzed photographs. The tests were performed in four biological replications with three technical repetitions constituting different observation fields of the examined well (n = 12). Biofilm levels were compared among the six culture conditions using all two-sided pairwise Wilcoxon rank-sum tests with Bonferroni correction (α = 0.05). All statistical analyses were performed in R (version 4.4.1).

Additionally, samples grown in E/Amix medium were subjected to in-depth analysis to determine the size of bacterial aggregates formed under these conditions. Aggregate sizes were normalized relative to the dimensions of the observation field in each analyzed image. Bacterial clusters occupying more than 1%, between 1% and 0.5%, between 0.5% and 0.1%, and less than 0.1% of the observation field were classified as large, big, medium, and small, respectively. Any measurements smaller than the size of a single bacterial cell were considered artifacts and excluded from the final analysis. After categorizing all aggregates in the image, the total area covered by each size category was calculated and expressed as a percentage of the overall aggregate area. To minimize the background fluorescence level, a 10% increase in the contrast was equally applied to all analyzed photographs. The analysis was performed in four biological replications with three technical repetitions constituting different observation fields of the examined well (n = 12).

### 4.3. Intestinal-Supporting Experiments

#### 4.3.1. Cell Line Tested

The epithelial normal colonocyte line CCD841 CoN (CRL-1790TM) was purchased from the ATCC collection (Manassas, VA, USA). EMEM10 (for details, see Section 4.1.4) was used for its multiplication. Incubation was carried out in an incubator (ThermoScientific, Waltham, MA, USA) at 37 °C in an atmosphere of 5% CO_2_.

#### 4.3.2. Experimental Evaluation of the Culture Conditions

To determine the optimal conditions for culturing the investigated colonocytes, the effects of FCS (at concentrations of 2.5%, 5% and 10%) and FBS (at concentrations of 10% and 20%) were evaluated. Additionally, the impact of ACF on the viability and adhesive capacity of the examined cell line was assessed. Consequently, the E/Amix medium (for details see Section 4.1.4) was selected for further tests using CCD841 CoN cells.

#### 4.3.3. Cytotoxicity Assay

A cytotoxicity assay for all three compounds was initially performed on 96-well adherent microplates following Ganot et al. (2013) [115]. CCD841 CoN cells were seeded at a density of 2 × 10^4^ cells/well and incubated for 24 h in EMEM10. Subsequently, wells were treated with the tested compounds at concentrations of 2 × MIC, MIC, and ½ × MIC for an additional 24 h. After that, cells were washed and treated with a medium containing 0.5 mg/mL of MTT (3-(4,5-dimethylthiazol-2-yl)-2,5-diphenyl-2H-tetrazolium bromide). After a 2-h incubation at 37 °C with 5% CO_2_, formazan crystals were dissolved using DMSO. The reaction was stopped by adding a Sorensen’s buffer, and absorbance was measured at 560 nm using a microplate reader (ASYS UVM340 BIOCHROM LTD., Cambridge, UK). The OD values directly reflected the metabolic activity of viable eukaryotic cells. Negative and positive controls included untreated CCD841 CoN cells and cells treated 2% SDS (sodium dodecyl sulfate, Merck, Darmstadt, Germany), respectively. Cell viability was expressed as a percentage relative to the negative control (100%). All assays were conducted in four independent replicates (n = 4).

The cytotoxicity test for BBR was additionally extended using semi-permeable inserts (pore size 1.0 μm, diameter 6.5 mm; Merck, Darmstadt, Germany) under conditions consistent with those applied in the following experimental procedures. CCD841 CoN cells at a density of 2 × 10^5^ cells/mL were seeded onto the apical side of the membrane (from the basal side was exposed to E10) and incubated for 24 h. After this, the basal medium was replaced with fresh EMEM10, while the BBR was applied to the apical side at concentrations of 2 × MIC, MIC, and ½ × MIC in E/Amix medium. After 24 h of incubation, semi-permeable inserts were transferred to a new 48-well plate, washed with PBS (Biowest Nuaillé, France), and MTT (0.5 mg/mL) was added to the apical side. Following incubation under standard conditions, first DMSO and then a Sorensen’s buffer were added. The resulting solution was transferred to a 96-well plate and absorbance was measured at 560 nm. The remaining steps were performed as described above.

Within each compound (BBR, CHE, SAN), dose effects were tested using two-sided pairwise Wilcoxon rank-sum tests with Bonferroni correction (α = 0.05). To compare compounds pooled across concentrations, two-sided pairwise Wilcoxon rank-sum tests with Bonferroni correction were applied across the three compound pairs. All statistical analyses were performed in R.

#### 4.3.4. Permeability Assay

To assess permeability, 4 × 10^4^ CCD841 CoN colonocytes per insert were seeded on the apical side of a semi-permeable culture insert. The inserts were placed into a 24-well plate with the basal side in constant contact with fresh EMEM10 medium. Cells were incubated for 3 days to promote monolayer formation. Cells used in the permeability assay were from passages 6 to 15.

Afterward, the basal medium was replaced with fresh EMEM10, and the apical side was supplemented with E/Amix medium enriched as follows: MIC of BBR (“BBR”), post-culture samples from a 72-h culture of *C. jejuni* of 10^8^ CFU/mL (“Camp”), or a combination of both (“BBRCamp”). Inserts were incubated for 24 h, 48 h, 72 h, 96 h, and 168 h. For each point, a control consisting only of CCD841 CoN cells in E/Amix medium was included (“Control”). All semi-permeable inserts remained in contact with EMEM10 medium on the basal side for the entire incubation period.

After the respective time points, media from both sides were removed. Next, the membranes and wells were washed with HBSS (Hanks’ balanced salt solution; General Chemistry Laboratory IITD PAN, Wrocław, Poland) pre-warmed to 37 °C. A barrier integrity assay using Lucifer Yellow (LY) was then performed according to the manufacturer’s protocol and based on the method of Pires et al. [116]. Pre-warmed HBSS was added to the basal side, and LY (100 μg/mL) was applied to the apical side of semi-permeable inserts. The systems were incubated for 3 h at 37 °C in 5% CO_2_ (the optimal incubation time was determined in a pilot study evaluating durations of: 30 min, 60 min, 90 min, 2 h, 3 h, 4 h, and 6 h).

After incubation, 100 μL of fluid from the basal side was collected and analyzed by fluorescence spectroscopy using a microplate reader Tecan Infinite M200PRO (Männedorf, Switzerland) at excitation/emission of 485 nm/535 nm. The percentage of LY that permeated through the membrane was calculated using a standard curve prepared for each experiment. The experiment was performed in independent replication (n = 7).

At each time point (24 h, 48 h, 72 h, 96 h), group differences among control, BBR, BBRCamp, and Camp were assessed with two-sided pairwise Wilcoxon rank-sum tests and Bonferroni correction across the six group pairs at that time point (α = 0.05). Prior to testing, outliers were screened within each Group × Time using the IQR rule (values < Q1 − 1.5 × IQR or >Q3 + 1.5 × IQR were excluded); no imputation was performed. All statistical analyses were performed in R.

#### 4.3.5. Morphostructural State Analysis

After designated incubation periods, the medium from the upper chamber of semi-permeable inserts was discarded and the semi-permeable membranes were subjected to fluorescent microscopic examinations. The membranes coming from the inserts were cut out with a scalpel and placed on glass slides. Then, each of them was stained with 20 µL of a PBS solution with a mix of two fluorescent dyes: FM 1–43 (0.5% *v*/*v*; ThermoFisher, Waltham, MA, USA) and DAPI (1% *v*/*v*; ThermoFisher, Waltham, MA, USA) to stain cell membranes and nuclei, respectively. Examinations were performed using a Carl Zeiss inverted fluorescence microscope. To minimize the background fluorescence level, a 10% increase in the contrast was equally applied to all analyzed photographs. The degree of area occupation by cell lines was calculated using Bioflux Montage software. The tests were performed in four biological replications with three technical repetitions constituting different observation fields of the examined well (n = 12). For all images obtained for cells cultured for 96 h, additional analysis of single-cell sizes was performed. Analyses were performed using the ImageJ software version 1.54j. The measurements were performed in three biological replications (n = 3).

#### 4.3.6. Re-Localization of Occludin

CCD841 CoN colonocytes seeded on inserts, collected at the 96-h time point, were stained with rabbit polyclonal anti-occludin antibodies conjugated with fluorescent dye CoraLite^®^594 (ThermoScientific, Waltham, MA, USA) at a dilution of 1:250. The effect of the supernatant coming from a 4-day *C. jejuni* culture on occludin location within the cell line was compared between BBR-treated and non-treated CCD841 CoN. Observations were made after 10 min of incubation in the dark using an inverted Carl Zeiss fluorescence microscope (GmbH, Jena, Germany). To minimize the background fluorescence level, a 10% increase in the contrast was equally applied to all analyzed photographs.

#### 4.3.7. Secretory Profile Analysis

Semi-permeable inserts for the metabolic state analysis were prepared as described in Section 4.3.4. with the only change being the use of EMEM medium without phenol red. Fluid samples in the volume of 100 μL were collected from the basal side of the inserts at three time points: 0 h, 48 h, and 96 h. These samples were promptly transferred onto ice and centrifuged (5000 rpm for 10 min at 4 °C). The pellet was separated from the supernatants under cold conditions, and the latter were immediately frozen at −80 °C and stored until analysis.

Sample Preparation for LC-MS/MS Analysis: The samples were thawed at 8 °C. A volume of 35 µL of an isotopically labeled amino acid mixture was added to 100 µL of culture medium. The samples were then mixed for 2 min at 1000 rpm at 8 °C. Subsequently, 900 µL of a precipitation mixture consisting of 75% ACN, 25% MeOH, and 0.2% FA was added. The mixture was vortexed for 2 min at 1200 rpm at 8 °C and then centrifuged for 6 min at 14,000 rcf at 4 °C. The resulting supernatant was transferred into chromatographic vials and subjected to LC-MS/MS analysis.

LC-MS/MS Analysis Conditions for Metabolomics: Chromatographic separations were performed using a Waters Acquity UPLC I-Class system equipped with an Acquity UPLC BEH Amide analytical column (150 × 2.1 mm, 1.7 µm particle size) coupled with a BEH Amide VanGuard pre-column (5 × 2.1 mm, 1.7 µm; Waters, USA). The column temperature was maintained at 50 °C throughout the analysis. Samples were kept at 9 °C in the autosampler prior to injection, with an injection volume of 5 µL per run.

Mass spectrometric detection was carried out on a SYNAPT G2 Si QTOF mass spectrometer (Waters, Milford, MA, USA), operating in both positive (ES^+^) and negative (ES^−^) electrospray ionization (ESI) modes. Data acquisition was performed using a data-independent acquisition (DIA) method. The ion source parameters were optimized for each polarity. For ES^+^ mode, the capillary voltage was set to 3 kV, the sampling cone to 30 V, and the source offset to 50 V. For ES^−^ mode, these values were adjusted to 2.5 kV, 40 V, and 30 V, respectively. The source temperature was maintained at 140 °C in both modes. Desolvation was conducted at 450 °C, with a desolvation gas flow rate of 900 L/h and a cone gas flow rate of 50 L/h. Collision energy ramping was applied from 20.0 to 40.0 eV in both ionization modes.

Chromatographic separation was achieved using a gradient elution program with a constant flow rate of 0.3 mL/min. The gradient began with 100% mobile phase B, which was held for the first 2 min. It was then linearly decreased to 70% at 7.7 min, followed by 40% at 9.5 min, and 30% at 10.25 min. From 13.0 min onward, the gradient was returned to 100% mobile phase B and held isocratically until 17.0 min for re-equilibration. A curve type of 6 was applied at all gradient transitions to ensure consistent flow modulation.

The mobile phases used in both positive and negative ion modes consisted of (A) acetonitrile:water (5:95, *v*/*v*) supplemented with 10 mM ammonium formate and 0.1% formic acid and (B) acetonitrile:water (95:5, *v*/*v*) containing the same additives. The combination of BEH Amide chemistry and optimized gradient conditions enabled efficient separation of polar metabolites, including amino acids and other small molecules. The experiment was performed in independent replication (n = 3).

Quality Control and Data Processing: Quality control (QC) samples, prepared from pooled aliquots of all extracts, were injected at the beginning of each analytical batch (eight injections for system conditioning) and subsequently after every nine sample injections to monitor system stability. A solvent/procedural blank was injected at the start of each run. Data acquisition was carried out in both positive and negative electrospray ionization (ESI) modes, with spectra processed separately. Following acquisition, raw data were centroided and lock-mass corrected using MSConvert v3.0, converted to mzML format, and then imported into MS-DIAL v4.92. Within MS-DIAL, feature detection, MS/MS deconvolution, spectral annotation using fragmentation libraries (including adduct handling), chromatographic alignment, and gap filling were performed. Metabolic features were defined as unique *m*/*z*–retention time pairs and quantified within MS-DIAL. Putative metabolite identities were assigned based on characteristic MS/MS fragmentation patterns matched against the spectral libraries.

## 5. Conclusions

Collectively, this study provides novel insights into the antibacterial, anti-biofilm, and barrier-protective properties of BBR against *C. jejuni*. For the first time, the MIC of pure BBR against *C. jejuni* was determined, confirming its antibacterial activity at 64 μg/mL. Our in vitro model of campylobacteriosis confirmed BBR is able to limit the pathological changes in colonocytes not only through direct antibacterial and anti-biofilm activity against *C. jejuni*, but also through a protective effect on the morphostructure and metabolic profile of colonocytes exposed to toxic components of this pathogen. These effects suggest that BBR may interfere with pathogen survival and host–pathogen metabolic interactions, while enhancing the host protective response.

Despite these promising results, the therapeutic use of BBR is limited by the low oral bioavailability and extensive first-pass metabolism. Modern drug delivery strategies—such as nanocarriers, liposomal formulations, and microfluidic technologies—offer potential solutions to enhance their absorption, stability, and intracellular distribution. Notably, lactoferrin-based nanoparticles co-loaded with BBR and other antimicrobials have shown improved antibacterial efficacy against intracellular and biofilm-forming pathogens.

Overall, our findings suggest that BBR is a multifunctional phytochemical with potential applications in preventing and treating *C. jejuni*-induced gastrointestinal infections and epithelial barrier dysfunctions. Further in vivo and clinical studies are required to confirm its efficacy and optimize the delivery system for future therapeutic use. They should focus on organ-on-a-chip and liver–gut flow models to overcome current bioavailability barriers.

## Figures and Tables

**Figure 1 ijms-26-10634-f001:**
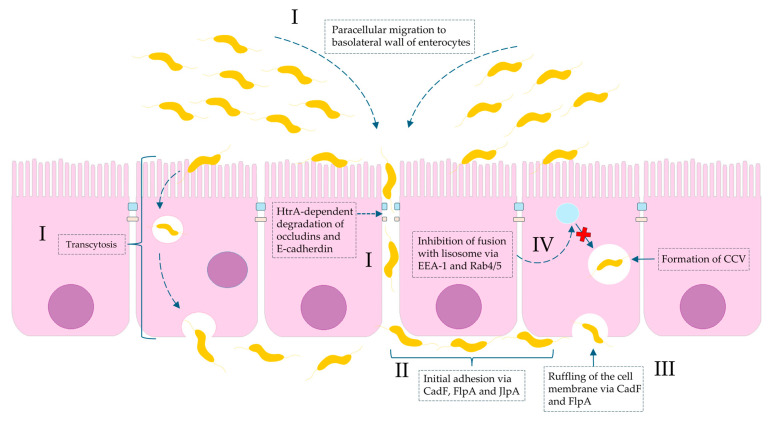
**Molecular basis of *Campylobacter* spp. invasion of intestinal mucosa with regard to virulence factors.** (**I**): invasion is a consequence of both paracellular cell migration to the basolateral side of the enterocyte via modifying fibronectin molecules and degradation of tight junctions (occludins and claudins) by HtrA protease and transcytosis (transfer from apical to basal part of enterocyte). (**II**): The initial adhesion of *Campylobacter* spp. to outer membrane proteins (OMPs) of the basolateral membrane of intestinal cells is due to bacterial proteins: CadF (fibronectin-binding protein), FlpA (fibronectin-like protein A), and JlpA (jejuni lipoprotein A-binding protein Hsp90). (**III**): The interaction of CadF and FlpA with fibronectin induces the remodeling of the actin cytoskeleton and leads to ruffling of the cell membrane. (**IV**): The intercellular survival of *Campylobacter* spp. cells inside infected cells are a result of the formation so-called CCV (*Campylobacter*-containing vacuole). The molecular interference with EEA-1 (early endosomal agent 1) and GTPases Rab4/5 leads to avoidance of vacuole-lysosome fusion.

**Figure 2 ijms-26-10634-f002:**
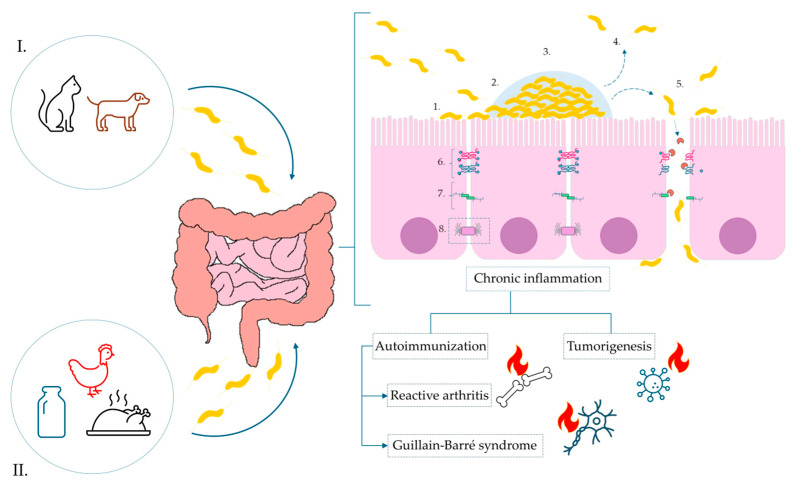
**Pathogenesis of *C. jejuni* infection and the impact of biofilm formation on the surface of the intestinal epithelium tight junctions.** Sources of infection marked, respectively, as: (**I**). animal reservoir-pet excrement, e.g., dogs and cats, (**II**). food products: contaminated dairy products, poultry meat. Infection: 1—Initial adhesion via surface proteins; 2—activation of biofilm formation; 3—maturation of biofilm; 4—activation of dispersion mechanisms; 5—degradation of tight junctions and E-cadherins via synthesis of secretory protease HtrA, paracellular transport of bacterial cells and migration to the basolateral membrane of enterocytes; 6—tight junctions; 7—adherens junctions; 8—desmosomes.

**Figure 3 ijms-26-10634-f003:**
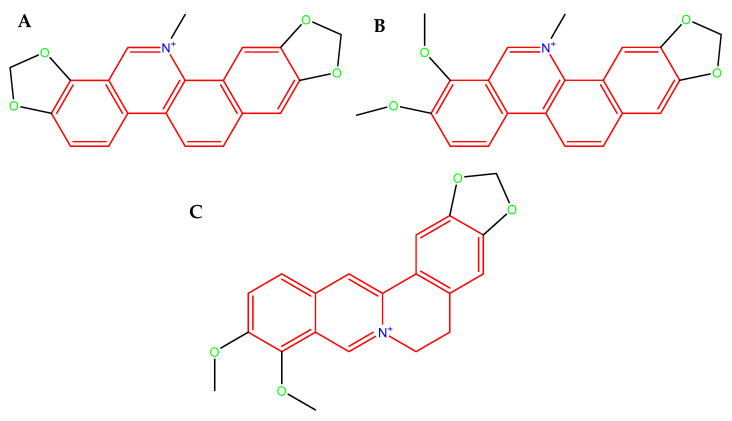
**Molecular structure of SAN.** (**A**): an orange-red benzophenanthridine alkaloid occurring as a chloride or sulfate salt, poorly soluble in water, readily soluble in organic solvents, fluorescent (λem ≈ 568 nm, strongly DNA-binding in its iminium ion form, molecular weight 367.8 g/mol (as free base) [59]; CHE (**B**): a yellow-orange crystalline benzophenanthridine alkaloid, typically occurring as a chloride salt, poorly soluble in water, but readily soluble in organic solvent, fluorescent (λem ≈ 560–570 nm), strongly binds to nucleic acids and proteins through its planar iminium ion form, molecular weight 348.4 g/mol (as free base) [60], and BBR (**C**): a yellow isoquinoline alkaloid that crystallizes as a chloride or sulfate salt, moderately soluble in water and highly soluble in organic solvents, fluorescent (λem ≈ 520–530 nm), capable strong interactions with nucleic acids and proteins via its planar quaternary ammonium structure, molecular weight 336.4 g/mol (as free base) [61].

**Figure 4 ijms-26-10634-f004:**
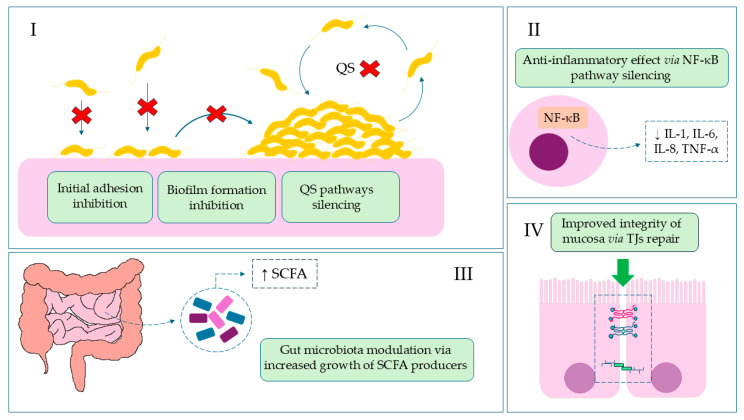
**The role of BBR in invasive gastrointestinal infections.** (**I**): Reduction of colonization, the first stage of infection, by blocking adhesive particles and stabilizing the epithelial barrier, which leads to inhibition of biofilm formation, quorum sensing (QS), and DNA synthesis; Red Cross—location of action/inhibition. (**II**): Reduction of inflammation and diarrhea symptoms by inhibiting NF-κB and lowering pro-inflammatory cytokine levels; (**III**): Restoration of intestinal balance by modulating the microbiota and increasing SCFA (short-chain fatty acids) bacteria; (**IV**): Reduction of systemic symptoms by strengthening the intestinal barrier.

**Figure 5 ijms-26-10634-f005:**
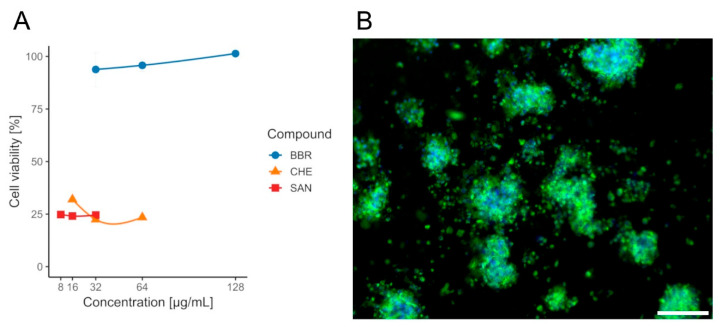
**Viability of CCD841 CoN cells after 24 h exposure to isoquinoline alkaloids measured by the MTT assay.** (**A**)—bars/points show mean ± SD (n = 4 per concentration). Concentrations: BBR 32/64/128 µg/mL (blue), CHE 16/32/64 µg/mL (orange), SAN 8/16/32 µg/mL (red). Within each compound, differences among concentrations were not significant. BBR yielded higher viability than CHE and SAN, whereas CHE vs. SAN were not different. (**B**)—the effect of BBR on colonocytes. Sample stained with FM 1–43 (green) and DAPI (blue) to visualize cell membranes and nuclei, respectively. Scale bar, 80 µm.

**Figure 6 ijms-26-10634-f006:**
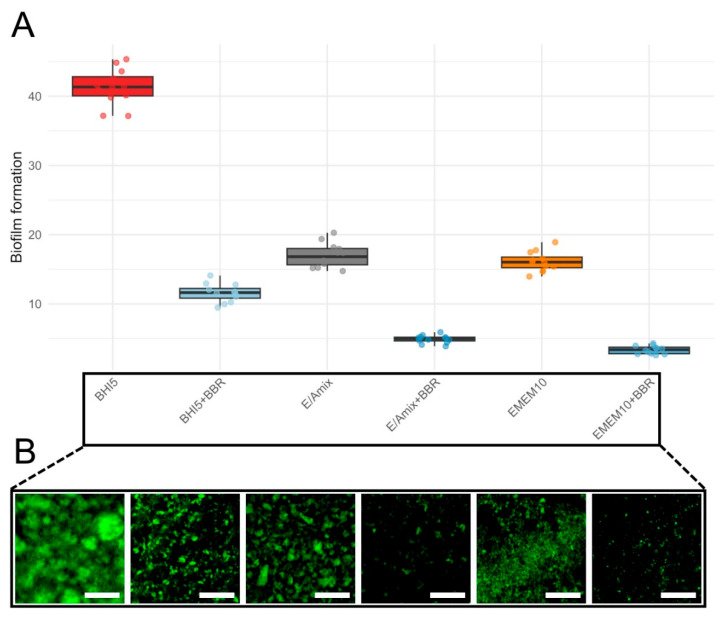
**Biofilm formation across culture conditions.** (**A**)—boxplots show the distribution of biofilm levels for six conditions: BHI5 (red), BHI5 + BBR (light blue), EMEM10 (orange), EMEM10 + BBR (medium blue), E/Amix (grey), and E/Amix + BBR (dark blue). Boxes indicate the IQR, center line the median, whiskers 1.5 × IQR; points are individual measurements. (**B**)—effect of BBR on biofilm formation under the experimental conditions corresponding to those shown in panel (**A**). Samples were stained with rabbit polyclonal anti-*C. jejuni* antibodies conjugated to FITC (dilution 1:10). Scale bar, 60 μm.

**Figure 7 ijms-26-10634-f007:**
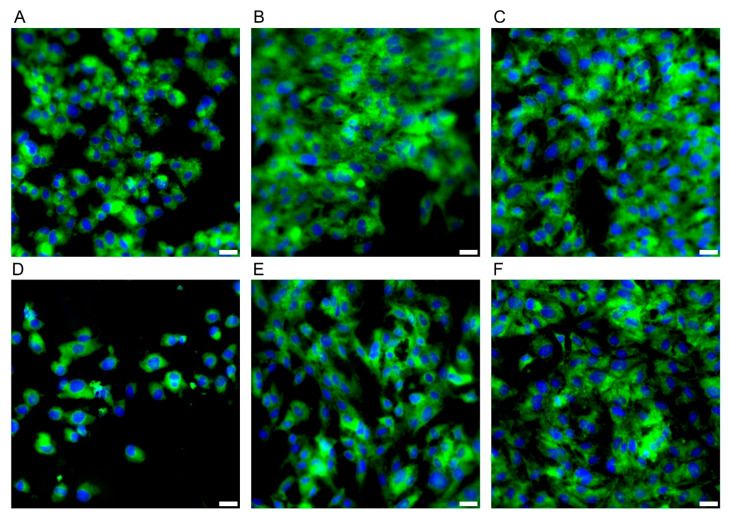
**Morphology of the CCD841 CoN cell line.** Panels (**A**–**C**) present cell growth in E/Amix medium after 24 h, 48 h, and 96 h, respectively, while (**D**–**F**) show cell growth in EMEM10 medium after 24 h, 48 h, and 96 h, respectively. Sample stained with FM 1–43 (green) and DAPI (blue) to visualize the cell membrane and nucleus, respectively Scale bars, 20 µm.

**Figure 8 ijms-26-10634-f008:**
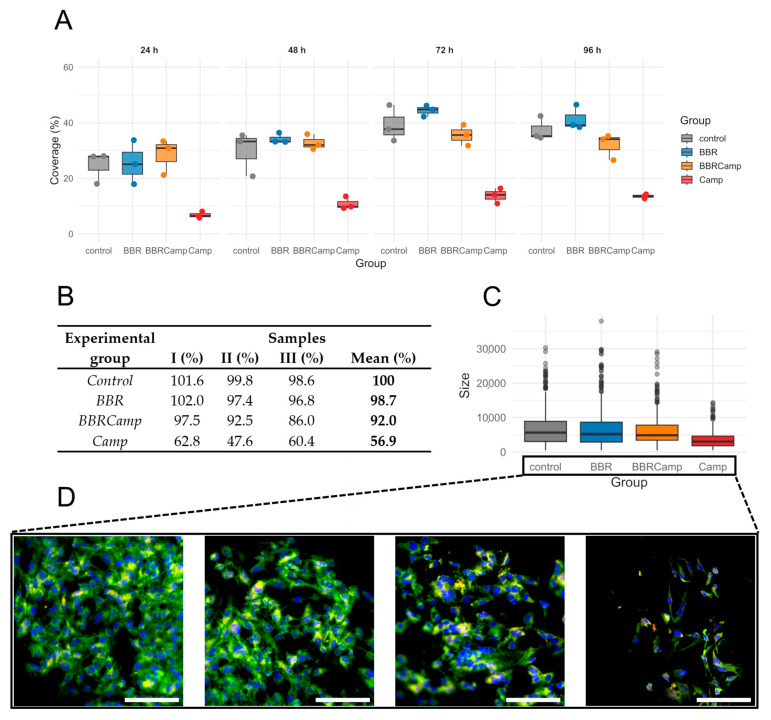
**Epithelial monolayer surface coverage (%) over time under four conditions.** (**A**)—boxplots with overlaid points show the distribution of measurements at 24 h, 48 h, 72 h, and 96 h for control (grey), BBR (blue), BBRCamp (orange; BBR + *C. jejuni* supernatant), and Camp (red; *C. jejuni* supernatant). Each point is an individual observation; boxes indicate the interquartile range (IQR), the center line is the median, and whiskers extend to 1.5 × IQR. The *y*-axis (0–60%) is shared across panels. (**B**)—table comparing epithelial monolayer surface coverage (%) from three repetitions at the time point of 96 h. (**C**)—boxplots display the distribution of cell sizes for the four experimental groups. Dots represent individual outliers and other plot elements are as described for panel (**A**). (**D**)—Morphology change of the tested colonocyte line and appearance of the monolayer under experimental conditions corresponding to, from left to right: control, BBR, BBRCamp (BBR + *C. jejuni* supernatant), and Camp (*C. jejuni* supernatant). Samples were stained with FM 1–43 (green), DAPI (blue) and anti-ocludin antibodies (red) to visualize the cell membrane and nucleus, respectively. Scale bar, 120 µm.

**Figure 9 ijms-26-10634-f009:**
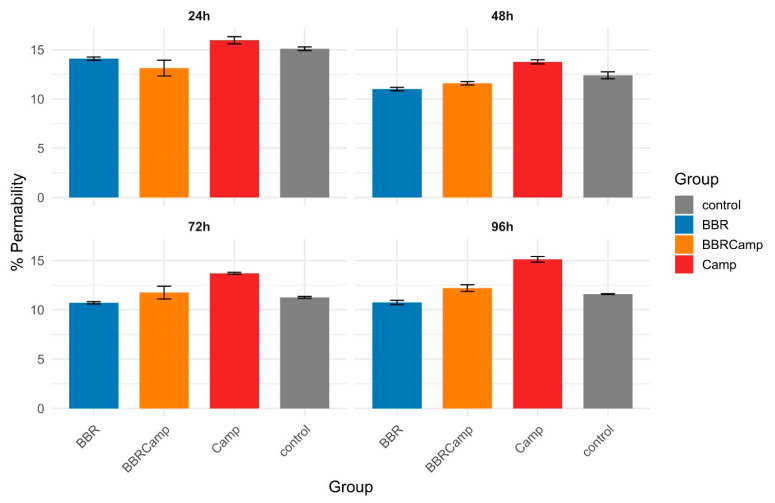
**Barrier permeability of CCD841 CoN monolayers over time.** Bars show the mean percentage of Lucifer Yellow (LY) passage (proxy of permeability) and error bars denote standard deviation (SD) for four conditions: control (grey), BBR (blue), BBRCamp (orange; BBR + *C. jejuni* post-culture supernatant), and Camp (red; *C. jejuni* supernatant alone). Panels display 24 h, 48 h, 72 h, and 96 h time points.

**Figure 10 ijms-26-10634-f010:**
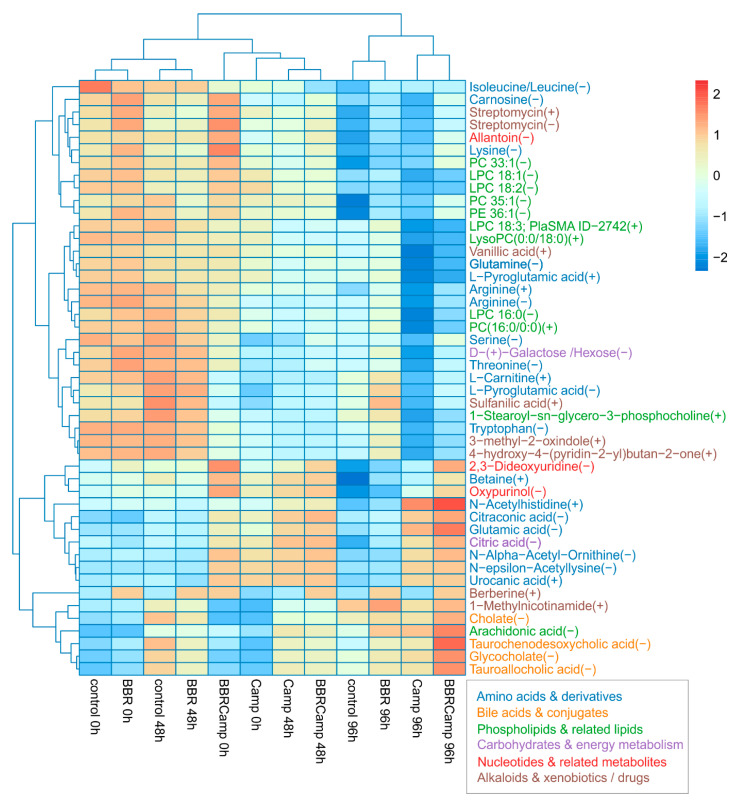
**Heat map of 47 metabolites that changed significantly across all groups.** Values are protein-normalized, log_2_-transformed means, and are row-scaled (Z-scores). Rows and columns are hierarchically clustered (Euclidean distance). Warmer colors indicate higher relative abundance and cooler colors lower. Columns are ordered by condition and time (control, BBR, Camp, BBRCamp at 0, 48, and 96 h). The maps reveal a clear structure by condition/time, with Camp at later times separating most strongly, while BBR and BBRCamp tend to cluster closer to the control.

**Table 1 ijms-26-10634-t001:** Anti-*C. jejuni* activity of CHE, BBR, and SAN under selected experimental conditions.

Condition	MIC [µg/mL]		
	CHE	BBR	SAN
**BHI**	32	32	8
**BHI5**	32	64	16
**ACF10**	32	64	16
**selected MIC**	**32**	**64**	**16**

**Table 2 ijms-26-10634-t002:** Microscopic analysis of the morphostructure of biofilm aggregates formed in experimental conditions.

Conditions	Size of Aggregates [%]			
	Small	Medium	Big	Large
**E/Amix**	40.8	46.5	9.0	3.7
**E/Amix + BBR**	87.2	12.8	0	0

Bacterial clusters occupying more than 1%, between 1% and 0.5%, between 0.5% and 0.1%, and less than 0.1% of the observation field were classified as large, big, medium, and small, respectively.

## Data Availability

The original contributions presented in this study are included in the article/Appendix A. Further inquiries can be directed to the corresponding author.

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
