# Peer review of "Impact of Isoquinoline Alkaloids on the Intestinal Barrier in a Colonic Model of *Campylobacter jejuni* Infection"

_ijms, 2025, doi:10.3390/ijms262110634_

Round 1

Reviewer 1 Report

Comments and Suggestions for Authors

Manuscript title:Impact of isoquinoline alkaloids on the intestinal barrier in a colonic model of Campylobacter jejuni infection

ID: ijms-3948422 / Type: Original research

Journal: International Journal of Molecular Sciences

--------------------------------------------------------------------------------------------------------

Dear Editors and Authors, thank you for inviting me to assess this manuscript’s suitability for publication in this critical journal, the International Journal of Molecular Sciences. Phytotherapy is, indeed, a growing field in modern medicine due to its various health benefits, often low prices, and minor adverse effects. However, research is still needed to validate findings and advocate for future clinical trials to pave the way for phytomedicine from the bench to the bedside. This original article delves into the reasoning behind the effects of isoquinoline alkaloids on the intestinal barrier in a colonic model of Campylobacter jejuni infection using preclinical models. This manuscript has its value, and I think it should be considered for publication after the Authors address MAJOR REVISIONS.

ABSTRACT

  1. The abstract should be rewritten entirely and structured using headings: “Background/Objectives,” “Methods,” “Results/Discussion,” and “Conclusions.”
  2. Numerical results alongside the respective p-values must be added to the abstract to convey the necessary results and strengthen the manuscript’s presentation.
  3. There are too many keywords. Please rephrase them using MESH Terms as indicators. Additionally, please do not use abbreviations in the abstract, including MIC and BBR.

INTRODUCTION

  1. Critical epidemiological data must be gathered in the introduction regarding the prevalence, morbidity, and mortality of invasive gastrointestinal infections among populations of all continents.
  2. The sanguinarine, chelerythrine, and berberine molecules should be studied in the introduction. Include their structures in a separate figure and bring to the caption key physicochemical information from PubChem. Additionally, carry information about conformational changes when hitting targets.
  3. A table summarizing the key virulence and morphological factors of the bacteria studied must be placed in the introduction section to pave the way for the complete exploration of berberine against this infection in the following sections. Additionally, it should be noted that the activities of berberine against these infections/bacteria may also be introduced in a separate figure or table. Make sure that adequate references are cited. Finally, a histological photo of the bacteria/infection would enhance the manuscript’s quality.

MATERIALS AND METHODS

  1. There is an inconsistency between the use of “FBS” and “FCS” in the text. While both terms refer to the same substance, consistency is essential. You may choose one term and use it throughout the manuscript. It would be better to organize the terminology of your manuscript better before the manuscript is resubmitted for re-evaluation by the reviewers.
  2. The purity of the tested compounds is mentioned, but no HPLC conditions are specified. While the supplier indicates purity, it would strengthen the methodology to include details about the analytical conditions for HPLC. This can be particularly important for reproducibility. Please ensure that this information is retrieved from the supplier.
  3. Provide more details about how growth was assessed in media optimization experiments and include exact compositions of all media.
  4. Report the final concentration of DMSO used in the cytotoxicity assays and whether it was tested for cytotoxicity at the given concentration in more detail in your methodology section.
  5. Clarify how the images were processed before quantification of area coverage and cell size. Were any image preprocessing steps used? It is essential to detail all this information in the methods to ensure that your study is reproducible.
  6. Include in more detail information about the inclusion of quality control samples (blanks, standards) in the LC-MS/MS analysis to ensure the robustness of the data.

RESULTS

  1. Your results should be structured using subsection headings. Please ensure that all information is disposed of within subsections to ensure that the results are easy to find and read in your manuscript.

DISCUSSION

  1. Pharmacodynamics and pharmacokinetics of each studied compound must be discussed in separate subsections within the discussion section. Toxicity and safety of each compound must also be a subsection of the discussion section based on the previously published studies in this regard.
  2. A subsection that delves into the limitations of this present manuscript and the future research endeavors based on your findings can also be included in the discussion section. This subsection must discuss the aspects mentioned above in more detail. Possible clinical implications can also be added to this subsection.

CONCLUSIONS

  1. Repeat the most prominent results here, including numerical results and their respective significance levels. In addition, a figure containing these results would be greatly appreciated. This figure can also include the characteristics of the study design, since a timeline for your study would greatly enrich the understanding of your experiments.

Thank you for your corrections. I am looking forward to hearing from you shortly.

Best.

Author Response

Dear Reviewer,

We would like to sincerely thank the Reviewer for their insightful and constructive comments, which have helped us to further improve the clarity of our manuscript. Below we present our response to the Reviewer's comments point by point:

ABSTRACT

  1. The abstract should be rewritten entirely and structured using headings: “Background/Objectives,” “Methods,” “Results/Discussion,” and “Conclusions.”

We would like to thank the reviewer for this comment. We would like to inform you that due to the complexity of our research, we are unable to write an abstract organized according to the recommended headings, as we would not be able to fit within the required 200 words. The very extensive methodological section and the results greatly exceed the word limit. Therefore, we have left the Abstract in its previous form. We have improved it slightly, and in its current form it reads as follows:

„Phytotherapy is a growing field of modern medicine, offering natural alternatives with multidirectional pharmacological effects. Among plant-derived bioactive compounds, isoquinoline alkaloids exhibit antioxidant, anti-inflammatory, and antimicrobial properties. Our in vitro model of campylobacteriosis confirmed that berberine reduces pathological changes in colonocytes not only through its direct antibacterial (minimum inhibitory concentration for pure berberine against Campylobacter jejuni was 64 μg/ml) and anti-biofilm (fourfold reduction in C. jejuni biomass) effects, but also through its protective effect on the morphostructure and secretory profile of host cells exposed to bacterial components. Furthermore, berberine stabilized intercellular junction proteins, modulated bile acid and arachidonic acid metabolism, and supported host-protective signaling pathways. These findings indicate that berberine acts through a dual mechanism—directly reducing bacterial virulence while enhancing intestinal barrier integrity and metabolic homeostasis. In summary, berberine appears to be multifunctional phytochemical in the development of new strategies for the prevention and treatment of C. jejuni-induced gastrointestinal infections and epithelial barrier dysfunction. The protective effect we have demonstrated may contribute to alleviating the phenomenon of “leaky gut,” commonly associated with campylobacteriosis.”

  1. Numerical results alongside the respective p-values must be added to the abstract to convey the necessary results and strengthen the manuscript’s presentation. Additionally, please do not use abbreviations in the abstract, including MIC and BBR.

We would like to thank the Reviewer for this comment. The most important figures are provided in the Abstract. We have omitted the rest so as not to unnecessarily complicate the message for the reader and to encourage them to read this Manuscript. At the same time, we would like to inform you that we have removed all abbreviations used in the Abstract, i.e., BBR and MIC. We hope that the Abstract in its current form will meet with the Reviewer's satisfaction and approval.

  1. There are too many keywords. Please rephrase them using MESH Terms as indicators.

We would like to thank you very much for your attention. We have reduced the number of keywords to 6. In addition, we have corrected them to comply with MeSH terminology. For the order's sake, they have been listed in alphabetical order.

INTRODUCTION

  1. Critical epidemiological data must be gathered in the introduction regarding the prevalence, morbidity, and mortality of invasive gastrointestinal infections among populations of all continents.

We would like to thank the reviewer for this valuable comment. Although epidemiological data on invasive infections are not available, we have cited data related to infections caused by contaminated food. It can be assumed that fatal cases are associated with those infections that are more life-threatening, i.e., invasive infections. The fragment we have added reads as follows:

„According to data from the World Health Organization (WHO), food-borne diseases remain a major global health problem. It is estimated that every year, nearly one in ten people worldwide fall ill due to unsafe food, with as many as 420,000 dying, including 125,000 children under the age of five [1]. Among these cases, IGIs, due to its pathomechanism of infection, are a particularly serious subgroup of intestinal diseases [2–5]”.

  1. The sanguinarine, chelerythrine, and berberine molecules should be studied in the introduction. Include their structures in a separate figure and bring to the caption key physicochemical information from PubChem. Additionally, carry information about conformational changes when hitting targets.

Thank you very much for pointing out the lack of this information. We have added molecular formulas of the compounds we studied. They are presented in Figure 2. In addition, below the figure, we have provided key physicochemical information and information on conformational changes. We invite the Reviewer to study this figure and its description in the latest version of our Manuscript.

  1. A table summarizing the key virulence and morphological factors of the bacteria studied must be placed in the introduction section to pave the way for the complete exploration of berberine against this infection in the following sections.

Thank you very much for this insightful comment. We have added a more detailed description of the virulence factors of C. jejuni in the Introduction. It has the following wording:

„Among Campylobacter strains, a crucial role of flagellin-related genes (e.g., flaA, flaB, flaC, flag, fliA, fliS, flhA) in the initiation of early adhesion and maintenance of the biofilm structure was demonstrated [31]. Additionally, genes involved in shaping the biofilm composition (e.g., waaF, lgtF and EptC) [21] and stabilizing its matrix (e.g., spot, ppk1/2 – stringent response regulators) [32] have been identified. A range of antioxidant genes, including, e.g., ahpC, katA and sodB, have also been shown to enhance biofilm resilience under environmental stress [21]. Recent studies highlight the role of cadF not only in facilitating enterocyte invasion via activation of the MAPK/ERK (microtubule-associated protein kinase/extracellular regulated kinase) signaling pathway [33], but also in promoting initial adhesion during biofilm formation [32]. Taking this into account, the biofilm created by C. jejuni, on the one hand, enables the persistence of this bacterium on the surface of the intestinal epithelium, while on the other hand, enables for the long-lasting, destructive deposition of lytic factors that negatively affect the condition of this organ [34]. C. jejuni has been shown to be involved in each stage of biofilm formation: initiation of early adhesion [31], composition modelling [21] and matrix stabilization [32]. In addition, a number of antioxidative proteins by C. jejuni have been detected, all of which determine the biofilm’s tolerance to environmental stress and its stabilization [21].

Importantly, it has been proven that the consequences of C. jejuni infection can be very serious and life-threatening. They can contribute to colorectal carcinogenesis and metastasis. This oncogenic potential has been associated with strains expressing cytolethal distending toxin (CDT), which can activate: i) the JAK2 (Janus kinase 2)/STAT3 (signal transducer and activator of transcription 3)/MMP9 (matrix metalloproteinase-9) signaling axis implicated in the tumor development [27,28], and ii) glycogen synthase kinase 3 beta (GSK3b), a key regulator of cancer metastasis [29]. Moreover, although a direct causal association has not been definitively established, C. jejuni infections has been implicated in immunoproliferative small intestinal disease (IPSID), also known as alpha heavy chain disease. Studies have demonstrated that a strong mucosal IgA response induced by chronic C. jejuni infection can lead to persistent stimulation of the mucosal immune system and aberrant secretion of truncated α-heavy chains. This process is thought to contribute to the pathogenesis of mucosa-associated lymphoid tissue (MALT) lymphoma involving the small intestine [30]. The role of Campylobacter spp. in invasive gastrointestinal infections is shown in Figure 1.”

We have supplemented it with Fig. 1. We encourage the Reviewer to familiarize themselves with the revised version of our Manuscript.

  1. Additionally, it should be noted that the activities of berberine against these infections/bacteria may also be introduced in a separate figure or table. Make sure that adequate references are cited. Finally, a histological photo of the bacteria/infection would enhance the manuscript’s quality.

We would like to thank the reviewer for this comment. We have enriched our manuscript with a figure illustrating the activity of BBR against invasive gastrointestinal infections. We have shown the stages of infection that can be/are (based on the action against invasive pathogens) inhibited by the isoquinoline alkaloid we studied. We encourage the reviewer to familiarize themselves with Figure 4 and its description. We hope that this presentation of BBR activity will meet with the reviewer's approval and satisfaction.

Unfortunately, we do not have histological images. However, we plan to expand our research using an in vivo model in the near future.

MATERIALS AND METHODS

  1. There is an inconsistency between the use of “FBS” and “FCS” in the text. While both terms refer to the same substance, consistency is essential. You may choose one term and use it throughout the manuscript. It would be better to organize the terminology of your manuscript better before the manuscript is resubmitted for re-evaluation by the reviewers.

We would like to thank the reviewer very much for this comment. We would like to inform you that the abbreviations FBS and FCS are not synonymous.

FCS is serum from calves, young animals belonging to the bovine family. Based on our experience and numerous publications (10.1128/JB.184.15.4187-4196.2002; 10.1556/EUJMI-D-15-00003), we cultivate Campylobacter strains in our laboratory in BHI medium enriched with FCS.

FBS, on the other hand, is general bovine serum (from both young and adult animals) and is dedicated to the cultivation of the CCD841 CoN colonocyte line (https://www.atcc.org/products/crl-1790). This cell line, like other intestinal lines, does not grow properly if the medium is enriched with FCS (we tested this in the laboratory while performing various experiments to facilitate the methodology). Similarly, the growth of the Campylobacter jejuni strain was significantly reduced in BHI medium with the addition of FBS.

Therefore, it was necessary to use two separate sera in the research model, taking into account the requirements of the tested strain and the tested cell line.  

  1. The purity of the tested compounds is mentioned, but no HPLC conditions are specified. While the supplier indicates purity, it would strengthen the methodology to include details about the analytical conditions for HPLC. This can be particularly important for reproducibility. Please ensure that this information is retrieved from the supplier.

We would like to thank the reviewer for this comment. We are attaching the protocols for the compounds used in the experiment. We trusted the manufacturer's claim that the purity was as stated in the documentation, as all the compounds tested were stored under the conditions recommended by the manufacturer. Therefore, there was no possibility of contamination.

  1. Provide more details about how growth was assessed in media optimization experiments and include exact compositions of all media.

We would like to thank the Reviewer for this comment. The compositions of the media used in the experiments are provided in the Supplementary Information. The media used for the growth of Campylobacter jejuni are listed in Tables S1 and S2. The medium dedicated to the growth of the tested colonocyte line is listed in Table S3. Table S4 contains information on the intestinal fluid. Unfortunately, the composition of this medium is proprietary to the manufacturer. Therefore, we have only provided the information that is available on the website.

  1. Report the final concentration of DMSO used in the cytotoxicity assays and whether it was tested for cytotoxicity at the given concentration in more detail in your methodology section.

We would like to thank the reviewer for this insightful comment. We would like to inform you that the concentrations of the tested compounds were prepared in such a way as to keep the DMSO concentration as low as possible. First, we prepared a stock solution, then diluted it in appropriate media and added it to colonocyte cells. According to our calculations, the compound prepared in this way contained 0.77% DMSO at the highest concentration of 512 μg/mL. This concentration did not affect the cells, which we ruled out at the beginning of our study (100% viability and unchanged morphology and characteristics, including adhesion to the medium). We did not include this information in the Materials and Methods section, as we considered it unnecessary. We assumed that our microscope images (BBR alone) indicate that the environment itself is not toxic to the cell line under study. If the Reviewer and Editors disagree, this information will be added to the appropriate place in the Manuscript.

  1. Clarify how the images were processed before quantification of area coverage and cell size. Were any image preprocessing steps used? It is essential to detail all this information in the methods to ensure that your study is reproducible.

We would like to thank the reviewer for this comment. We would like to inform you that before quantifying the surface coverage and cell size in the analyzed images, the contrast was increased by 10%. We have added the following information: “To minimize a background fluorescence level, a 10% increase in the contrast was equally applied to all analyzed photographs” to the Materials and Methods section in the appropriate places.

  1. Include in more detail information about the inclusion of quality control samples (blanks, standards) in the LC-MS/MS analysis to ensure the robustness of the data.

We would like to thank the reviewer for this comment. We have added an additional section providing detailed information on the inclusion of quality control samples in the LC-MS/MS analysis. It reads as follows:

“Quality Control and Data Processing: Quality control (QC) samples, prepared from pooled aliquots of all extracts, were injected at the beginning of each analytical batch (eight injections for system conditioning) and subsequently after every nine sample injections to monitor system stability. A solvent/procedural blank was injected at the start of each run. Data acquisition was carried out in both positive and negative electrospray ionization (ESI) modes, with spectra processed separately. Following acquisition, raw data were centroided and lock-mass corrected using MSConvert v3.0, converted to mzML format, and then imported into MS-DIAL v4.92. Within MS-DIAL, feature detection, MS/MS deconvolution, spectral annotation using fragmentation libraries (including adduct handling), chromatographic alignment, and gap filling were performed. Metabolic features were defined as unique m/z–retention time pairs and quantified within MS-DIAL. Putative metabolite identities were assigned based on characteristic MS/MS fragmentation patterns matched against the spectral libraries.”

RESULTS

  1. Your results should be structured using subsection headings. Please ensure that all information is disposed of within subsections to ensure that the results are easy to find and read in your manuscript.

We fully acknowledge the Reviewer’s concern regarding the structure of the Results section. In the original version of the manuscript, this section was divided into several subsections. However, upon careful consideration, we found that such a structure made the text less cohesive and more difficult to follow. The subsections were often to short (in some cases consisting of only a few sentences), and the information presented in tchem appeared fragmented, which disrupted the logical flow of the results. By merging these subsections, we aimed to provide a more continuous and comprehensible narrative of our findings, while also avoid unnecessary repetition. Given that our study was designed as an integrated and logically coherent whole, we believe that this approach most effectively reflects the nature or our results and enhances the readability of the manuscript. We sincerely hope, tha the Reviewer will understand our reasoning and agree that this revision contributes to the overall clarity consistency of this article.

DISCUSSION

  1. Pharmacodynamics and pharmacokinetics of each studied compound must be discussed in separate subsections within the discussion section. Toxicity and safety of each compound must also be a subsection of the discussion section based on the previously published studies in this regard.

A subsection that delves into the limitations of this present manuscript and the future research endeavors based on your findings can also be included in the discussion section. This subsection must discuss the aspects mentioned above in more detail. Possible clinical implications can also be added to this subsection.

We would like to thank the Reviewer for pointing out the points that need to be addressed in the “Discussion” section. As suggested by the Reviewer, we have added a paragraph on pharmacokinetics and pharmacodynamics. However, we focused only on BBR, as it was the main focus of our research. SAN and CHE showed strong cytotoxicity towards the colonocyte line we studied, so describing their pharmacokinetics and pharmacodynamics in this Manuscript would be beyond our intended purpose. We combined this paragraph with the toxicity of BBR and its poor bioavailability, and expanded it to include future research projects that may contribute to solving this problem. The fragment we added reads as follows:

„Despite the promising results obtained in this study, it is important to consider the pharmacokinetic and pharmacodynamics limitations of BBR. Following oral administration, BBR exhibits low bioavailability, due to extensive first-pass metabolism. Zuo et al. (2006) demonstrated that the liver is primary site of BBR biotransformation with major metabolites and glucuronide conjugates detected in hepatic tissue and bite within 0.5 and 1 hour after administration, respectively [106]. Although the gut microbiota of germ-free rats (administrated 40 mg/kg BBR) showed limited direct metabolic activity toward BBR, it significantly influenced the enterohepatic circulation of metabolites [106].

Therefore, the development of alternative delivery strategies and methods of administration is crucial for improve BBR absorption and therapeutic efficacy. Nanocarriers, liposomal formulations, and microfluidic systems have shown encouraging results on improving BBR solubility, stability, and intestinal absorption [107–109]. Furthermore, structural modifications of BBR have improved pharmacological performance, while reducing adverse effects [110,111]. Recent studies have demonstrated that lactoferrin-based nanoparticles loaded with BBR and SAN, combined with conventional antibiotics, significantly enhance intracellular drug delivery and antibacterial activity compared to free drug forms [112]. This highlights the potential of nanocarrier-mediated targeting system to overcome intracellular sequestration of pathogens and improve the therapeutic outcomes of poorly bioavailable alkaloids. Future research employing flow-based pharmacokinetic models, including hepatic cell systems and organ-on-a-chip technologies, will be crucial for elucidating BBR metabolism, toxicity, and tissue accumulation [113,114]. These approaches will support the establishment of standardized dosing, optimized delivery routes, and defined treatment durations for clinical use. The present findings provide new insights into the antimicrobial, anti-biofilm and intestinal barrier-modulating potential of BBR and open avenues for future in vivo investigations.”

CONCLUSIONS

  1. Repeat the most prominent results here, including numerical results and their respective significance levels. In addition, a figure containing these results would be greatly appreciated. This figure can also include the characteristics of the study design, since a timeline for your study would greatly enrich the understanding of your experiments.

We would like to thank the reviewer for pointing out the poorly constructed summary. We have rewritten it and added the most important results and future plans based on our findings. We have also taken into account the limitations of the BBR study, which should be considered in future in vivo studies. We have decided not to include the Figures, as adding 3 Figures to the Introduction significantly increased the volume of our Manuscript. We hope that the Reviewer will understand our point of view. In the latest version of the Manuscript, the “Conclusion” section reads as follows: 

„Collectively, this study provides novel insights into the antibacterial, anti-biofilm, and barrier-protective properties of BBR against C. jejuni. For the first time, the MIC of pure BBR against C. jejuni was determined, confirming its antibacterial activity at 64 μg/mL. Our in vitro model of campylobacteriosis confirmed BBR is able to limit the pathological changes in colonocytes not only through direct antibacterial and anti-biofilm activity against C. jejuni, but also through a protective effect on the morphostructure and metabolic profile of colonocytes exposed to toxic components of this pathogen. These effects suggest that BBR may interfere with the pathogen survival and host-pathogen metabolic interactions, while enhancing the host protective response.

Despite these promising results, the therapeutic use of BBR is limited by the low oral bioavailability and extensive first-pass metabolism. Modern drug delivery strategies – such as nanocarriers, liposomal formulations, and microfluidic technologies – offer potential solutions to enhance its absorption, stability, and intracellular distribution. Notably, lactoferrin-based nanoparticles co-loaded with BBR and other antimicrobials have shown improved antibacterial efficacy against intracellular and biofilm-forming pathogens.

Overall, our findings suggest that BBR is a multifunctional phytochemical with potential applications in preventing and treating C. jejuni-induced gastrointestinal infections and epithelial barrier dysfunctions. Further in vivo and clinical studies are required to confirm its efficacy and optimize delivery system for future therapeutic use. It should focus on organ-on-a-chip and liver-gut flow models to overcome current bioavailability barriers.”

Dear Reviewer, we would like to thank you very much for your insightful review of our manuscript. We appreciate the time you have spent improving our article. We are sending you a revised version in accordance with your recommendations. We have put a lot of effort into it. We hope that you will find it satisfactory and acceptable. We encourage you to familiarize yourself with the improved version of our Manuscript.

Yours sincerely,

Anna Duda-Madej

Reviewer 2 Report

Comments and Suggestions for Authors

The well-structured study “Impact of isoquinoline alkaloids on the intestinal barrier in a colonic model of Campylobacter jejuni infection” addresses a highly relevant topic within the field of activities of plant alkaloids.

a very important in vitro model, including not only the study of antibacterial and anti-biofilm activity against C. jejuni, but also a protective effect of isoquinoline alkaloids from Ch. majus L. on the morphostructure and metabolic profile of colonocytes exposed to toxic components of this pathogen was presented. Importantly, this complex investigation allow to found that alkaloid berberine possessed not only antibacterial but also have host protective mechanisms. The multipotential activity of this compound positioned it as a promising natural agent in future therapeutic strategies aimed at treating campylobacteriosis infections and relieving the pathological consequences of intestinal barrier damage.

The writing is clear, and the results are presented in a way that makes the paper both engaging and easy to follow.

 Overall, this is a valuable contribution to the literature, and I consider the work highly publishable in its current form, with only minor adjustments if the authors wish to further strengthen the discussion.

Author Response

Dear Reviewer,

We are delighted that you fully appreciate our manuscript. We are very pleased that the presentation of our results is transparent and clear. The study was multidirectional and highly complex, and compiling the results in a way that is clear to the reader was quite a challenge. In order to meet your expectations and those of the second reviewer, we have improved our “Discussion” section. We have added a paragraph in which we briefly describe the pharmacokinetics and pharmacodynamics of berberine. We also address the issue of the bioavailability of this compound and indicate directions for future research, taking into account our results, which may help to overcome this problem. The added paragraph reads as follows:

“Despite the promising results obtained in this study, it is important to consider the pharmacokinetic and pharmacodynamics limitations of BBR. Following oral administration, BBR exhibits low bioavailability due to extensive firs-pass metabolism. Zuo et al. (2006) demonstrated that the liver is primary site of BBR biotransformation with major metabolites and glucuronide conjugates detected in hepatic tissue and bite within 0.5 and 1 hour after administration, respectively [106]. Although the gut microbiota of germ-free rats (administrated 40 mg/kg BBR) showed limited direct metabolic activity toward BBR, it significantly influenced the enterohepatic circulation of metabolites [106].

Therefore, the development of alternative delivery strategies and methods of administration is crucial for improve BBR absorption and therapeutic efficacy. Nanocarriers, liposomal formulations, and microfluidic systems have shown encouraging results on improving BBR solubility, stability, and intestinal absorption [107–109]. Furthermore, structural modifications of BBR have improved pharmacological performance while reducing adverse effects [110,111]. Recent studies have demonstrated that lactoferrin-based nanoparticles loaded with BBR and SAN, combined with conventional antibiotics, significantly enhance intracellular drug delivery and antibacterial activity compared to free drug forms [112]. This highlights the potential of nanocarrier-mediated targeting system to overcome intracellular sequestration of pathogens and improve the therapeutic outcomes of poorly bioavailable alkaloids. Future research employing flow-based pharmacokinetic models including hepatic cell systems and organ-on-a-chip technologies will be crucial for elucidating BBR metabolism, toxicity, and tissue accumulation [113,114]. These approaches will support the establishment of standardized dosing, optimized delivery routes, and defined treatment durations for clinical use. The present findings provide new insights into the antimicrobial, anti-biofilm and intestinal barrier-modulating potential of BBR and open avenues for future in vivo investigations.”

We also encourage you to read the improved version of our Manuscript.

Yours sincerely,

Anna Duda-Madej

Round 2

Reviewer 1 Report

Comments and Suggestions for Authors

Thank you for your corrections.

Comments on the Quality of English Language

The manuscript would be strengthened by a more in-depth review of the English language.